# Soft Finger Rehabilitation Exoskeleton of Biomimetic Dragonfly Abdominal Ventral Muscles: Center Tendon Pneumatic Bellows Actuator

**DOI:** 10.3390/biomimetics8080614

**Published:** 2023-12-15

**Authors:** Dehao Duanmu, Xiaodong Li, Wei Huang, Yong Hu

**Affiliations:** 1Orthopedics Center, The University of Hong Kong-Shenzhen Hospital, Shenzhen 518053, China; 2Department of Orthopaedics and Traumatology, The University of Hong Kong, Hong Kong, China; 3Department of Rehabilitation, The Second Affiliated Hospital of Guangzhou Medical University, Zhanjiang 524002, China

**Keywords:** soft actuator, central tendon-based, soft exoskeleton, muscles of dragonflies

## Abstract

The development of soft robotics owes much to the field of biomimetics, where soft actuators predominantly mimic the movement found in nature. In contrast to their rigid counterparts, soft robots offer superior safety and human–machine interaction comfort, particularly in medical applications. However, when it comes to the hand rehabilitation exoskeletons, the soft devices have been limited by size and material constraints, unable to provide sufficient tensile strength for patients with high muscle tension. In this paper, we drew inspiration from the muscle structure found in the tail of dragonflies and designed a novel central tendon-based bellows actuator. The experimental results demonstrated that the central tendon-based bellows actuator significantly outperforms conventional pneumatic bellows actuators in terms of mechanical output. The tensile strength of the central tendon-based bellows actuator exceeded that of pneumatic actuators more than tenfold, while adding only 2 g to the wearable weight. This finding suggests that the central tendon-based bellows actuator is exceptionally well-suited for applications demanding substantial pulling force, such as in the field of exoskeleton robotics. With tensile strength exceeding that of pneumatic bellows actuators, this biomimetic design opens new avenues for safer and more effective human–machine interaction, revolutionizing various sectors from healthcare to industrial automation.

## 1. Introduction

Traditional rigid robots drive structural motion through methods such as linkage motors, while soft robots employ soft actuators composed of gases, liquids, or soft solids to drive structural motion [1]. Many designs of soft actuators draw inspiration from principles in biomimicry, such as artificial muscles emulating human muscle motion [2] or trunk-like inflatable actuators inspired by the structure of elephant trunks [3]. Compared to rigid actuators, soft actuators demonstrate exceptional performance in adapting to irregular shapes and complex environments. Additionally, their softness reduces the risk of collisions or contacts. These adaptability and safety features have propelled the widespread adoption of soft robots in the field of healthcare.

Benefiting from the advantages, over the past few decades, soft robotics technology has made remarkable advancements in the field of wearable rehabilitative exoskeletons, particularly in hand function rehabilitation [4]. Hand exoskeletons pose unique challenges due to the small size of finger joints and the intricate nature of hand movements, necessitating stringent requirements in terms of device dimensions and degrees of freedom. Early rigid hand rehabilitation robots, such as MIT Manus [5] and Gloreha [6], while offering increased degrees of freedom, encountered limited adoption in hand rehabilitation due to their substantial weight and the potential for secondary injuries associated with rigid actuators. In recent years, a plethora of soft hand rehabilitation exoskeleton robots have emerged, including devices like HandSOME (The Catholic University of America, Washington, DC 20064, USA) [7] and Bioservo SEM Glove (Biosevo Technology AB, Stockholm, Sweden) [8]. These systems incorporate soft actuators capable of accommodating finger joint movements and delivering the required pushing and pulling forces for rehabilitation training. To alleviate the burden of device weight on patients, these soft hand rehabilitation exoskeleton robots typically employ pneumatic soft actuators to minimize their own weight.

Pneumatic soft actuators achieve motion or mechanical output by altering the shape of structures such as highly elastic material shells or bellows through the regulation of internal air pressure. However, owing to inherent material constraints, pneumatic soft actuators have limited load-bearing capabilities, and the tensile force they can generate is contingent upon their own shape and internal negative pressure [9,10]. In the context of hand rehabilitation robotics, the application of pneumatic soft actuators is restricted by their size and shape, making it challenging to produce sufficient tensile force to counteract muscle tension in patients with finger joint spasms [11].

In contrast to pneumatic soft actuators, cable-driven soft actuators operate on an entirely different principle. They induce motion in the actuator structure by contracting or extending cables, akin to the functioning of the human musculotendinous and skeletal system. When the cables are tightened, the required tensile force and motion are generated. Cable-driven actuators not only provide ample tensile force but also exhibit sufficient flexibility to adapt to various shapes and environments, facilitating multi-degree-of-freedom motion that closely mimics biological movement patterns. Consequently, they are highly suited for application in medical robotics, offering a more natural fit for human motion and proving particularly advantageous in the field of rehabilitation medical robots.

Considering the bidirectional mechanical output capabilities inherent to pneumatic soft actuators and the substantial tensile strength and flexibility of cable-driven counterparts, pneumatic soft actuators featuring cable-based structures prove highly suitable for integration into the design of lightweight wearable rehabilitative exoskeleton robots necessitating significant tensile forces and bidirectional force output. Examples such as TCA (Twisted-and-Coiled) actuators (Colorado State University, Fort Collins, CO, USA) [12] and the OctArm Continuum Robot (The Pennsylvania State University, United States) [13] effectively harness the combined principles of cables and pneumatics to drive their soft actuators. Nevertheless, in the context of cable-driven pneumatic actuators with external tethering, the tensile force is transformed into circumferential forces within the pneumatic actuator, demanding the use of multiple cables to maintain directional balance. In the realm of hand rehabilitation exoskeleton robots, space constraints along the exterior of the fingers pose challenges for accommodating externally tethered pneumatic actuators with intricate, voluminous structures. Moreover, the bending direction of the fingers remains fixed and hand function rehabilitation exoskeleton robots primarily necessitate a unidirectional bending motion.

The exploration of biomimicry in robotics and prosthetics has drawn inspiration from the remarkable biomechanics observed in various creatures [14], particularly the exquisite abilities of arthropods. In some studies of soft robots, arthropods such as Megarhyssa and lobsters have brought a lot of inspiration to researchers [15,16]. In past studies on bionic robots, the unique flight structure of dragonflies has been imitated by some researchers [17,18,19,20]. However, not much robot attention has been paid to the mechanism of the dragonfly belly curling phenomenon. As an arthropod, the structure of the dragonfly’s tail is like that of other arthropods [21]. Our inspiration comes from this arthropod structure: the principle of bending, straightening, and contraction of the arthropod through the anatomy of insects [22]. As shown in Figure 1a, during the water-dipping behavior of dragonflies, the abdomen of the dragonfly can autonomously bend and extend along specific directions, exhibiting similarities to the bending characteristics of finger joints. Based on the anatomical structure of the dragonfly abdomen presented in Figure 1b, this flexibility stems from the presence of ring-shaped muscles within the dragonfly abdomen, facilitating controlled bending through muscle contraction and relaxation. Additionally, analogous to tendons in human anatomy, elastic membranes or connecting tissues interlink the abdominal segments, storing and releasing energy to enhance bending motions. Furthermore, in some studies of insect anatomy, the structure of the abdomen of arthropods such as bees is observed through electron microscopy. Elastic membranes or connecting tissues exist between the abdominal segments of the dragonfly, resembling structures akin to tendons, which can store energy and enhance the bending motion of the abdomen [23].

In emulation of the abdominal bending motion observed in arthropods like dragonflies, we have designed a dual-drive soft actuator by integrating a cable-driven actuator within a central pneumatic bellows-based soft actuator, shown in Figure 1c, referred to as CCPB (central cable-driven pneumatic bellows). The working principle and structural design of this kind of actuator are inspired by the body structure of dragonflies or other arthropods [24]. The inflation or deflation of pneumatic muscles is similar to the contraction or relaxation of muscles, which can provide thrust or pulling force, while the pulling wire is equivalent to a reinforced intersegmental membrane, which can provide strong pulling force. When the pneumatic bellows actuator is pressurized, it generates thrust, and when the pneumatic bellows actuator is depressurized, or when the cable-driven actuator contracts, or when both actions occur concurrently, it can exert tensile forces. Utilizing relatively rigid cables, this dual-drive soft actuator with a central cable exhibits excellent tensile capabilities. When employed as the driving mechanism in hand rehabilitation exoskeletons, this CCPB actuator can not only generate forces for driving finger flexion and extension in both directions but can also deliver ample tensile forces to counteract excessive muscle tension. This innovative actuator design stands as a testament to the integration of biological inspiration into engineering solutions, offering advancements in soft robotics and rehabilitation technology through biomimetic design principles.

Inspired by the structural marvel of dragonfly tails, which is found in most arthropods [22], this novel actuator ingeniously employs cables to harness pulling forces beyond the conventional capabilities of pneumatic actuators. This innovative design not only emulates but also enhances the natural biomechanics observed in the tails of dragonflies, offering a remarkable solution that holds significant implications and inspiration for the application of soft robotics. By capitalizing on the insights drawn from the anatomical features of dragonfly tails, this actuator unveils new possibilities in soft robotics, transcending the limitations of traditional pneumatic actuators and paving the way for more efficient and versatile applications in the realm of robotics.

The main contribution of this article lies in the design of a central cable-driven pneumatic bellows actuator (CCPB), inspired by the bending phenomenon observed in arthropods such as dragonflies. This actuator has been specifically devised for application in soft exoskeleton robots for hand rehabilitation, with the dual objectives of providing bidirectional force outputs while generating sufficient tensile forces to counteract excessive muscle tension resulting from hand spasms. This soft actuator relies on the inflation and deflation of an external pneumatic bellows to exhibit basic stretching and contracting capabilities, enabling the generation of forces in two directions. Although the centrally positioned cable-driven actuator possesses a unidirectional stretching capability, allowing it to provide force in one direction, the inherent material properties of the cable-driven actuator enable it to withstand substantial tensile forces. By selecting high-torque motors, it becomes feasible to furnish the exoskeleton robot’s joints with adequate tensile force for extension, thereby complementing the bidirectional force outputs achieved by the pneumatic bellows actuator.

## 2. Materials and Methods

### 2.1. Design

The abdominal region of dragonflies exhibits a remarkable capability for unidirectional bending and extension. This unique ability stems from the contraction and relaxation of annular muscles located between each abdominal segment, facilitating both bending and extension (Figure 1b). The intersegmental muscles can provide push or pull force, and the intersegmental membrane can also be passively stretched. Furthermore, the presence of passive stretching in elastic membranes aids in storing energy, thereby accelerating the bending motion of the abdomen. When the dragonfly’s tail is straightened, the upper and lower muscles and the intersegmental membrane are in a stretched state. When the tail is bent, the intersegmental membrane is stretched, the upper muscles relax, and the lower muscles contract. In the case of the central cable-driven pneumatic bellows actuator, the peripheral pneumatic bellows section serves a role akin to the annular muscles in a dragonfly’s abdomen, capable of providing thrust or tension. Simultaneously, the central cable, while passively offering tensile resistance, can actively produce tension through motorized control. When the dual-drive muscle stretches, the pneumatic actuator inflates to provide thrust and the cable actuator stretches passively; when the dual-drive muscle contracts, there are many different working states, including the pneumatic actuator providing pulling force alone, the pneumatic actuator and the cable actuator providing the pulling force together, and the cable actuator providing the pulling force separately.

In the design process of the central cable-driven pneumatic bellows actuator, while emulating the structural features of a dragonfly’s abdomen, the peripheral pneumatic actuator component adopts a flattened, guided-bending bellows actuator (GBBA) [25]. This design diverges from the traditional cylindrical bellows actuator as it exhibits a preference for bending along the shorter axis, effectively converting linear pushing and pulling forces into torque. Previous experiments have yielded a maximum linear tensile force of approximately 15 N for this guided-bending bellows pneumatic actuator, attributed to the vacuum pump’s maximum negative pressure of around −80 kPa [26]. Additionally, in designs featuring corrugated tubes constructed from TPE material with a wall thickness of only 0.4 mm, the maximum tensile capacity is limited to approximately 18 N, exceeding which results in structural collapse [25].

The primary purpose of the central cable is to transfer the overall tensile load of the soft actuator from the bellows shell structure to the central cable section. The linear output of tension primarily relies on the tension applied to the cable-driven actuator by the servomotor.

### 2.2. Manufacture

Aligned with the design concept of the central cable-driven pneumatic bellows actuator, the entire soft actuator is divided into two main components: the external pneumatic guided-bending bellows actuator and the central cable-driven actuator. Correspondingly, each component has its respective driving mechanisms and the connecting and sealing components interfacing between the two systems, as shown in Figure 2. Consequently, the manufacturing process of the central cable-driven pneumatic bellows actuator is segregated into the fabrication of the pneumatic system, the cable-driven system, and the sealing and connecting components.

The pneumatic system of the central cable-driven pneumatic bellows actuator encompasses the pneumatic generation section, pneumatic transmission section, and the guided-bending bellows actuator [25]. To select suitable manufacturing methods and materials, it is essential to determine the fundamental physical parameters required by the guided-bending bellows in the hand rehabilitation exoskeleton context. Building upon our prior research, the guided-bending bellows actuator must deliver an elongation ratio of at least 1.5 to facilitate bending and extension at the finger joints of the exoskeleton [27]. Additionally, it should withstand pressures exceeding 20 N without collapsing to provide the necessary force for hand rehabilitation [25,28]. The manufacturing methods for the pneumatic soft actuator primarily consist of mold forming and additive manufacturing. Given that achieving a minimal wall thickness is imperative for bellows structures providing the elongation ratio at finger-sized dimensions, and considering the current resolution limitations of additive manufacturing methods for soft materials (approximately 1.2 mm), mold forming techniques such as blow molding or injection molding are chosen. Due to the challenges associated with processing soft materials in small-sized molds, blow molding is favored. Common materials used in blow molding include EVA or TPE. Given the non-cylindrical shape of the designed flattened structure of the guided-bending bellows actuator, TPE material is preferred for enhanced structural integrity and uniform blow molding [29], which can be seen in Figure 3a. The pneumatic generation section comprises a vacuum pump, a high-pressure pump, and electromagnetic valves. In our setup, we have adopted series-connected pumps serving as the vacuum pump and high-pressure pump, capable of delivering pressures of +120 kPa and −80 kPa, respectively. The electromagnetic valves employed in the system are SMC electromagnetic valves, characterized by a rapid response time of 5 ms.

The cable-driven actuator comprises the cable fixed end, cable, winding wheel, and servo motor. To achieve sufficient tensile capacity, a multi-strand steel wire with a diameter of 0.8 mm was selected for the cable within the central cable-driven pneumatic bellows flexible actuator. As shown in Figure 3b, the cable fixed end is bonded to the head of the pneumatic bellows and secured using an aluminum alloy ring. The winding wheel of the cable-driven actuator is fabricated using PLA material and 3D printing technology, with a diameter of 20 mm. In the manufacturing of our central cable-driven pneumatic bellows flexible actuator applied to hand rehabilitation exoskeletons, a 20 kg servo motor is employed. The output torque of the servo motor significantly exceeds the requirements of the application, while ensuring that its size and weight remain within acceptable limits.

The assembly of the central cable-driven pneumatic bellows actuator involves integrating the cable-driven actuator component within the guided-bending pneumatic bellows, thus combining the pneumatic transmission section with the cable-driven power section. The cable’s fixed end can be securely fastened to the head of the guided-bending pneumatic bellows using an aluminum alloy ring and sealed with adhesive. Given that there is no relative motion between these components, sealant considerations are not required in this context. The cable enters the pneumatic pathway through the tail end of the guided-bending pneumatic bellows and necessitates separation from the pneumatic pathway. In Figure 3b, this separation is accomplished by employing a Y-shaped three-way quick-connect joint, which guides the cable into a separate Teflon tubing. In this tubing, PA material is employed for sealing. To obtain good sealing, a hot melt glue gun is first used to heat the PA rod to about two hundred degrees Celsius to obtain an extremely fluid PA liquid, and then a syringe is used to inject the slightly cooled but still fluid PA liquid into the PU tube. The injection length is 1–2 cm. After the PA liquid in the tube cools and solidifies, the metal cable is heated and pulled out with force to clean the remaining PA solid on the metal cable. For the manufacturing and installation of dual-drive actuators for the entire finger, fixed connectors are installed at the front and rear of each pneumatic actuator. The purpose is to connect individual pneumatic actuators and to limit the position of the cable actuator. As shown in Figure 3c, the air path is connected between the two pneumatic actuators through an air pipe, and the cable is in the center of the air pipe. The end of the cable actuator passes through the positioning plate and is wound onto the turntable of the steering gear.

### 2.3. Design of Serial and Parallel CCPBs

When employed as the driver for a soft hand rehabilitation exoskeleton robot, a single CCPB actuator can only actuate one finger joint. Therefore, serial connection is required to drive the movement of an entire finger, like shown in Figure 4a. At least two or three serially connected central CCPB actuators are needed for the exoskeleton of a single finger. The entire hand exoskeleton shares a common pneumatic generation device, thus necessitating parallel usage for five serially connected CCPB actuators.

The serial design of CCPB actuators aims to interconnect multiple actuators to achieve the movement of an entire hand, as shown in Figure 4c. In this design, multiple CCPB actuators are serially connected in sequence, with each actuator corresponding to different joints of the finger. To maintain the airtight integrity of the serial assembly, it is essential to ensure that the end of each CCPB actuator is connected to the input of the next actuator while preserving the continuity of the cable. We have designed a hermetic sealing connector to ensure that gas can smoothly flow from one actuator to the next, as shown in Figure 3c. Additionally, at the connection points, Teflon tubing is used inside both guided-bending pneumatic bellows to serve as a conduit for the cable, preventing direct abrasion of the cable against the bellows.

In soft hand rehabilitation exoskeleton gloves, the serially connected CCPB actuators require a multi-way parallel design to ensure that each cable-driven actuator operates simultaneously and that gas is evenly distributed to each actuator [7,30]. As shown in Figure 4b, in the parallel pneumatic circuitry, we employ a one-to-five quick-release connector as an air distribution manifold to ensure that the positive or negative pressure generated by the pneumatic generation device is simultaneously connected to the pneumatic circuits of all five fingers. The connection at the junction between each actuator’s serial tubing and the quick-release connector is established using PU (Polyurethane) tubing and Y-shaped connectors. The additional pathway of each Y-shaped connector connects to Teflon tubing to separate the cable-driven system from the pneumatic system, employing piston devices to maintain airtightness. During the entire glove installation process, because the Y-shaped interface will become loose after repeated use, we designed a manifold integrating five Y-shaped quick-release openings based on the structural characteristics of the Y-shaped interface. The actual picture is shown in Figure 4d.

To guide the five cables from each actuator into the same cable winding pulley, we have designed a specialized guiding plate system. The guiding plate features multiple cable grooves, with each groove corresponding to a cable from one of the fingers. The primary function of the guiding plate is to direct the five cables from their respective actuators to the position of the cable winding pulley, ensuring smooth cable winding. To reduce the wearables’ weight, components such as servos, power supplies, and controllers are connected to the guiding plate via Teflon tubing, minimizing the overall weight of the hand rehabilitation exoskeleton.

Through this parallel design, the five serially connected cables are efficiently guided into a single cable winding pulley, facilitating coordinated motion. This design enables CCPB actuators to work synergistically within the hand rehabilitation exoskeleton, providing the wearer with the required tension and motion control while ensuring system stability.

### 2.4. Control

In contrast to independently controlling pneumatic actuators by adjusting air pressure or regulating cable-driven actuators by controlling cable tension, the coordinated motion of both the pneumatic and cable-driven components is pivotal during the operation of the central cable-driven pneumatic bellows flexible actuator (CCPB actuator). Depending on the specific output requirements, the CCPB actuator can assume three distinct operational states: pneumatic bellows actuator taking the initiative with cable-driven actuator in passive motion, cable-driven actuator taking the initiative with pneumatic bellows actuator in passive motion, and both cable-driven actuator and pneumatic bellows actuator jointly taking the initiative in output generation. Under these working modes, the working status of each module of the system is shown in the following table (Table 1).

To realize the flexion or extension of the finger exoskeleton and to prevent the interference of the two actuators, the motion states of each component of the dual-drive actuator are various in different working modes, and switching working modes requires cutting off the power source first. Then, he exoskeleton is returned to its original state.

In the control schematic, the pneumatic component control is achieved using a microcontroller to regulate the opening and closing of the vacuum pump, high-pressure pump, and electromagnetic valves (Figure 5). This control mechanism allows for the adjustment of air pressure and flow within the pneumatic circuit, thereby controlling the motion of the bellows. When the high-pressure pump and high-pressure valve are activated, while the vacuum pump, vacuum valve, and relief valve remain closed, pressure within the pneumatic circuit increases, causing the bellows to expand and generate thrust. Conversely, when the vacuum pump and vacuum valve are activated, with the high-pressure pump, high-pressure valve, and relief valve closed, pressure within the pneumatic circuit decreases, leading to the contraction of the bellows and the generation of tension. In situations where the high-pressure valve and vacuum valve are closed while the relief valve is open, the internal air pressure within the bellows aligns with the external atmospheric pressure. During this phase, the bellows can be passively compressed without causing structural damage.

The control of the cable-driven component primarily relies on servo motors to drive the winding drum for tension adjustment. The magnitude of the tension is determined by the diameter of the winding drum and the rotational speed of the servo motor. Given a servo motor angular velocity of *ω*, and with a defined winding drum radius of *r*, the linear velocity *ν* at the end of the central cable-driven pneumatic bellows flexible actuator (CCPB actuator) can be calculated as
(1)v=ω·r,

Additionally, the force *F* can be determined by
(2)$$F=\frac{M}{r}$$
where *M* represents the torque applied.

When the cable-driven actuator is controlled independently, it can achieve contraction of the entire actuator. In this scenario, the pneumatic component operates in a venting state, causing passive contraction of the pneumatic bellows. Controlling the servo motor’s rotational speed is sufficient to control the entire system.

Correspondingly, when the pneumatic component is controlled independently, adjusting the operating time of the high-pressure system or vacuum system is sufficient to regulate the pressure, denoted as ‘p’, within the pneumatic circuit, enabling continuous thrust or tension output from the bellows actuator. Due to the elastic nature of pneumatic actuators, there is a certain hysteresis in the relationship between end displacement and air pressure during deformation, which necessitates calibration through experiments.

When the pneumatic bellows and cable-driven actuator work together, the cable’s velocity should actively adapt to the pneumatic actuator’s movement speed to minimize structural interference resulting from relative motion. The servo motor’s velocity should be adjusted according to the formula:(3)$$\omega =\frac{S}{r\cdot t},$$
where *S* represents the end displacement of the pneumatic actuator, and *t* is the duration of a single stroke of the pneumatic actuator.

## 3. Experimental Setup

To assess the performance and advantages of the central cable-driven pneumatic bellows (CCPB) actuator, we constructed both a linear experimental setup and a bending experimental setup. These setups were designed to conduct separate tests, evaluating the mechanical and kinematic outputs of the CCPB actuator in the linear direction, as well as the torque and angular outputs when applied to finger joints. 

The experiments conducted in the linear direction provide a tangible measurement of the actuator’s tensile and compressive force outputs, as well as its displacement outputs. These measurements serve as the foundation for achieving torque and angular outputs when applied to rotating joints. By quantifying the tensile and compressive forces under different pneumatic pressures and servo motor movements, we can directly determine the forces generated by the actuator, thereby establishing its maximum load-bearing capacity. This is particularly crucial when considering the use of the central cable-driven pneumatic bellows (CCPB) actuator in soft hand rehabilitation exoskeletons, where patients with high muscle tension need to be accommodated. The experimental results aid in ensuring that the exoskeleton system operates effectively without exceeding its load-bearing limits, ensuring that its performance meets the required standards. Additionally, displacement output measurements play a pivotal role in calibrating the coordinated control of the pneumatic and cable-driven systems. In subsequent iterations of the actuator, these measurements can serve as reference parameters for designing cable system angular velocities and reel dimensions.

In the linear experiments, the CCPB actuator’s two ends of the central cable-driven pneumatic bellows were horizontally fixed on a test rig. The rig employed two pairs of parallel horizontal slides to maintain the actuator’s linear motion and counteract the effects of gravity, which are shown in Figure 6c. In the mechanical experiments, the actuator’s tail end was fixed, while its head end was connected to another fixed end through a force sensor, maintaining its original length. In the scenarios where the two actuator systems acted independently, when the pneumatic system was pressurized, the actuator generated thrust; when the pneumatic system was depressurized, the actuator produced tension. When the cable system’s servo motor rotated to tighten the cable, the actuator also exerted tension. In situations where both actuator systems worked in conjunction, when the pneumatic system was pressurized and the cable system’s servo motor remained stationary or the cable was relaxed, the actuator produced thrust. Conversely, when the pneumatic system was depressurized and the cable system’s servo motor rotated to tighten the cable, the actuator generated tension. In the kinematic experiments, the actuator’s tail end was fixed, while its head end was connected to a sliding plate, allowing it to slide freely with the assistance of horizontal slides. A laser displacement sensor scanned the position of the sliding plate to measure the actuator’s displacement. In cases where the two actuator systems operated independently, when the pneumatic system was pressurized, the actuator extended, and when the pneumatic system was depressurized, the actuator retracted. When the cable system’s servo motor rotated to tighten the cable, the actuator retracted, and when the servo motor rotated to relax the cable, the actuator returned to its original length. In scenarios where both actuator systems operated concurrently, when the pneumatic system was pressurized and the cable system’s servo motor was rotated to relax the cable, the actuator extended. When the pneumatic system was depressurized and the cable system’s servo motor rotated to tighten the cable, the actuator retracted. To prevent severe mechanical interference within the actuator, it was essential to ensure that the cable system’s servo motor was not in a state of rotation to tighten the cable when the pneumatic system was pressurized.

The experiments on rotational joints simulated the angular rotation data achievable when the central cable-driven pneumatic bellows (CCPB) actuator operated on finger joints. Angle sensors (SMD IMU sensor) were positioned at the rotational axis of the joint to precisely measure angular changes. The experiments gathered angle information under three different driving modes: pneumatic-only, cable-only, and combined pneumatic- and cable-driven. These tests aimed to capture the actuator’s performance in generating angular motion at finger joints, providing valuable data for the evaluation of its functionality in a rehabilitation exoskeleton context.

The dual-drive actuator used for the hand rehabilitation exoskeleton has different output characteristics in different working modes. A no-load experiment in a complete finger exoskeleton was designed to explore the different working modes of the robot. Three dual-drive actuators, connected by fasteners and connectors, form a rehabilitation exoskeleton for one finger. The glove base and finger cuffs printed from TPU material are glued to the exoskeleton as the wearable interactive part of the rehabilitation exoskeleton. By controlling the working conditions of the air pumps, solenoid valves, and servo motor, four motion states are executed in three different modes, including pneumatic-only drive, cable-only drive, and dual drive, in which the finger bending state only occurs under pneumatic-only drive.

## 4. Results

The status of a single finger’s dual-drive exoskeleton completing various actions under different drive modes is shown in Figure 7. In the initial state, since the length of the TPU glove base is shorter than the original length of the actuator, the exoskeleton is in a naturally curved state as shown in Figure 7a. In the pneumatic-only drive mode, the bellows actuator of the finger exoskeleton expands, driving the exoskeleton to flexion (Figure 7b); in the pneumatic-only drive mode, the cable-only drive mode and the dual-drive mode, the exoskeleton can contract and extension. The difference is that when the pneumatic is driven alone to straighten, the bellows can only be folded to drive the exoskeleton into a horizontal state, shown in Figure 7c. When the cable is driven alone to straighten, the bellows will be slightly buckled, shown in Figure 7d. Only in the dual-drive mode are the bellows completely folded, shown in Figure 7e. More specific mechanical parameters are shown below in Figure 8 in the CCPB actuator experiment tested separately.

Compared with other soft hand function rehabilitation robots, the CCPBs exoskeleton can provide greater pulling force in dual-drive mode, which is especially advantageous for the rehabilitation of patients with hand muscle spasms. And simply compared with the soft exoskeleton we studied previously that was driven solely by pneumatic actuators, the wearing weight of the dual-drive soft hand functional exoskeleton robot only increased by 15 g, but the pulling force provided increased by 90 N, which is more than five times the original pulling force.

The following table lists the features and applicability of different working modes. The pneumatic method is the drive method that must be provided in this exoskeleton that can achieve finger bending. Currently, most commercial soft hand function rehabilitation robots and various studies use this method because of its comfort in human–computer interaction. The pneumatic method is influenced by the effective area of the actuator and the negative pressure provided by the vacuum pump during finger extension. The maximum pulling force that can be achieved has limitations and is suitable for patients with muscle weakness but not suitable for patients with muscle spasm. In the cable-driven mode, the rehabilitation gloves can only achieve the action of extension of the fingers, but the power output is much greater than that in the pneumatic mode. In the dual-drive mode, it can not only achieve finger bending but also provide greater pulling force during finger extension listed in Table 2.

In the linear experiment, we tested the mechanical output performance of the CCPB actuator under pneumatic drive, cable drive, and common drive states over multiple cycles. As shown in Figure 8a, when the CCPB actuator is driven by pneumatics alone, the output force is like the mechanical output of an ordinary bellows. However, when the cable actuator is driven alone, the pulling force is a fixed value because the exact rotation angle is given. When the cable system is activated and intervenes in the pneumatic system, the thrust generated by the pneumatic actuator is minimal. Therefore, when in use, the thrust process requires a separate movement using a pneumatic actuator, while the pulling process can be generated by a wire actuator.

The purpose of the displacement experiment is to keep the movement speeds of the two close to each other and reduce structural damage caused by mechanical interference. Due to the soft characteristics of the bellows actuator, the speed of the central cable actuator can exceed that of the bellows actuator during the contraction process, as shown in Figure 8b. Because of the instability of pneumatic control, it is difficult for the two actuators to completely overlap.

In the bending experiment, we used the IMU (Inertial Measurement Unit) to read the angle changes of the cable and pneumatic system. It can be seen from Figure 8c that the cable actuator cannot output thrust, so it can only release the joint to the initial position. By reasonably adjusting the rotational speed of the servo, the speed at which the pneumatic actuator and the cable actuator pull the knuckle joint can be kept similar.

## 5. Discussion and Conclusions

In this study, we have successfully designed a central cable-driven pneumatic bellows (CCPB) actuator that combines the advantages of pneumatic bellows actuators and cable-driven actuators. It can achieve bidirectional force output and provide greater pulling force during finger extension. The design of the CCPB actuator draws inspiration from biological principles, mimicking the internal structural principles found in the abdomen of dragonflies, which include muscles and elastic membrane structures. It simulates the bidirectional output force of muscles in a single structure and the one-way enhancement of tensile force by the elastic membrane.

The airtightness of the CCPB actuator is effectively ensured using Y-shaped airway connections and piston devices. This feature is crucial for maintaining precise control and efficient operation in both pneumatic and cable-driven modes. Through the coordinated control of the pneumatic and cable-driven components, we have enhanced the tensile force output and the overall tensile capacity of the actuator while preserving its original bidirectional mechanical capabilities. This integrated approach empowers the CCPB actuator to generate significant tensile forces, rendering it not only suitable for patients with hand functional muscle weakness but highly suitable for applications where effectively managing elevated muscle tension, as frequently encountered in rehabilitation exoskeletons, is paramount [31].

In our experiments, we set up linear and joint-mimicking test setups to assess the mechanical and kinematic outputs of the CCPB actuator in both linear and rotational contexts and tested the movement characteristics of a single-finger exoskeleton under different drive modes. This provided us with precise measurements of the force and displacement characteristics of the CCPB actuator compared to traditional pneumatic bellows actuators, all under the same pneumatic input conditions. These experiments allowed us to quantify the CCPB actuator’s performance and better understand its capabilities in various scenarios.

Our experimental results demonstrate that the CCPB actuator outperforms traditional pneumatic bellows actuators in terms of force and displacement characteristics, especially in scenarios where high tensile forces are required to counteract excessive muscle tension.

The development and successful testing of the CCPB actuator represent a significant step forward in the field of flexible actuation systems. The integration of pneumatic and cable-driven components provides a promising solution for applications requiring versatile and robust force generation. Future work may focus on further optimizing the design and control strategies for the CCPB actuator to enhance its performance and broaden its applicability in areas such as robotics, prosthetics, and rehabilitation devices.

## Figures and Tables

**Figure 1 biomimetics-08-00614-f001:**
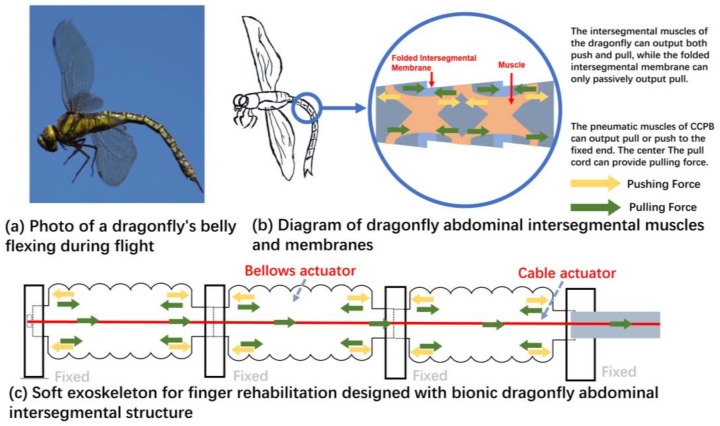
Inspired by the abdominal movement of dragonflies, the central pull cable bellows actuator. (**a**) Photograph of the bending motion of the tail during the dragonfly water-dipping behavior; (**b**) schematic illustration of the structure of the dragonfly tail; (**c**) a soft exoskeleton actuator for finger rehabilitation.

**Figure 2 biomimetics-08-00614-f002:**
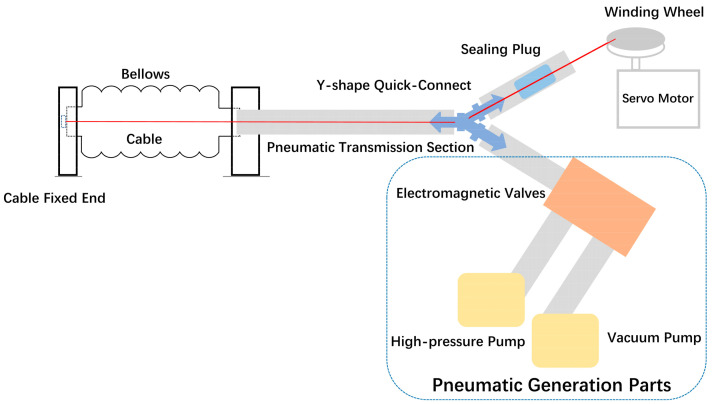
Schematic diagram of the structural principal design of the central cable bellows pneumatic and cable device.

**Figure 3 biomimetics-08-00614-f003:**
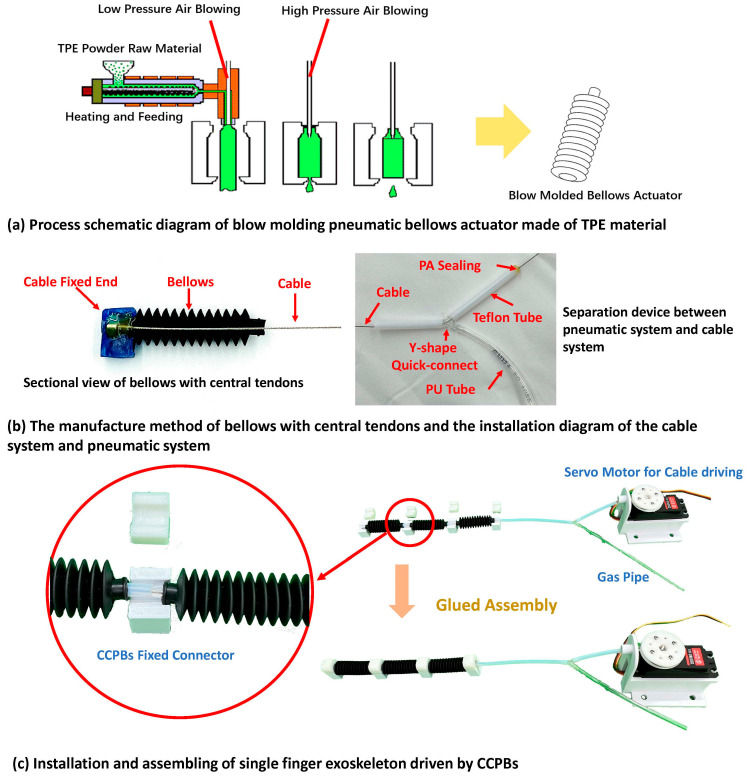
Manufacturing diagram of CCPB actuator.

**Figure 4 biomimetics-08-00614-f004:**
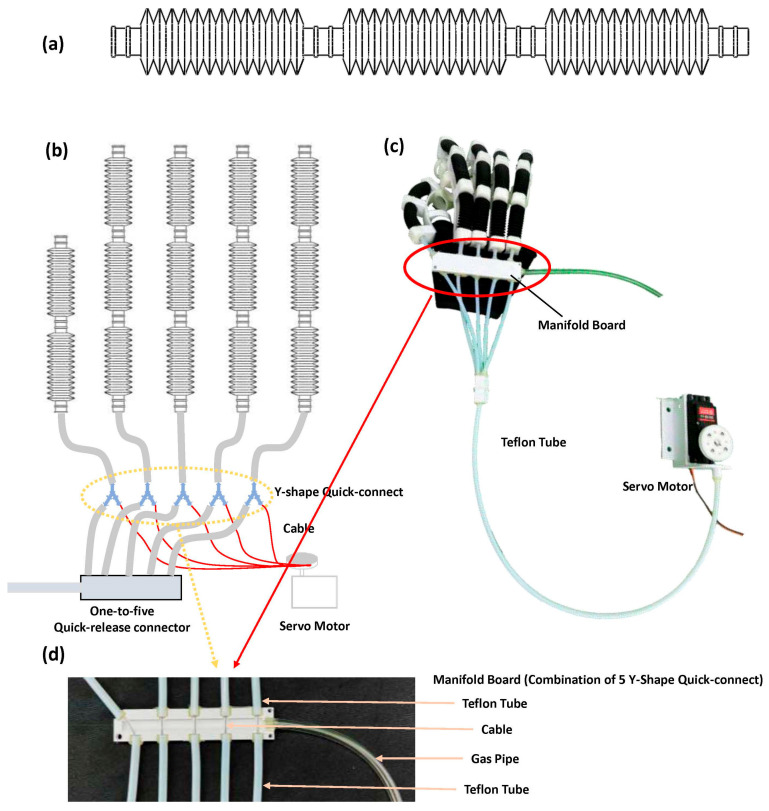
Schematic diagram of series and parallel structure of CCPB actuators. (**a**) Three CCPB actuators connected in series; (**b**) concept diagram of the pneumatic and cable system of five parallel-connected CCPB actuator series; (**c**) real photo of rehabilitation glove driven by parallel CCPBs series; (**d**) manifold board designed by combining 5 Y-type quick-release connectors for parallel connection of CCPBs.

**Figure 5 biomimetics-08-00614-f005:**
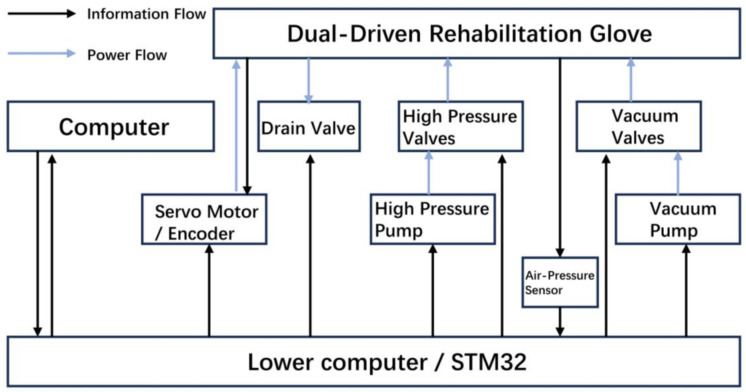
Control schematic of dual-driven rehabilitation glove.

**Figure 6 biomimetics-08-00614-f006:**
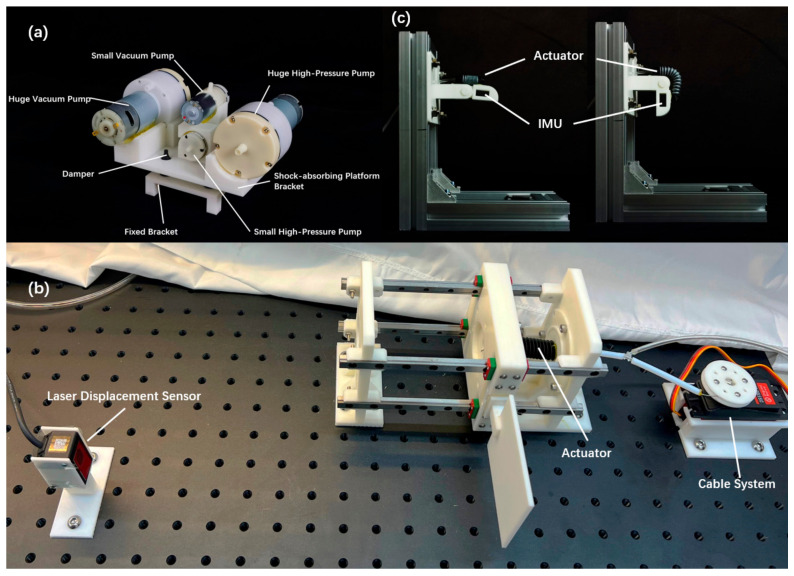
Experimental setup. (**a**) Double-stage parallel air pump effectively increases the air pressure and flow required in experiments; (**b**) linear experimental slide device; (**c**) bending experiment rotating joint.

**Figure 7 biomimetics-08-00614-f007:**
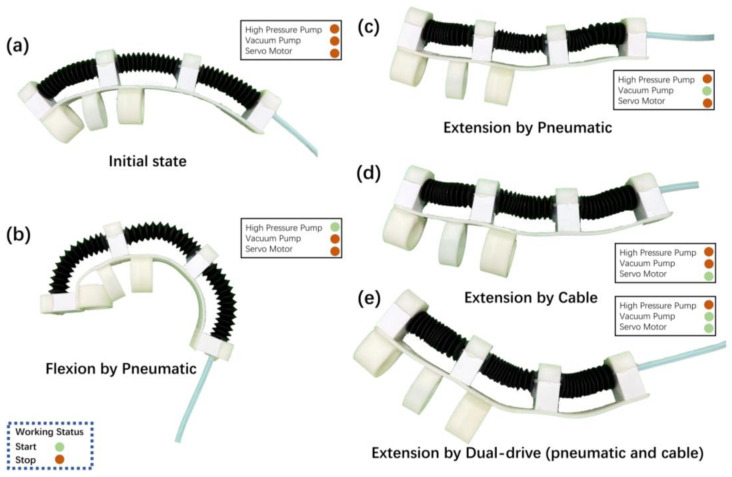
The status of the single-finger exoskeleton in different drive modes: (**a**) Original state; (**b**) Flexion state in inflation of pneumatic-only mode; (**c**) Extension state in deflation of pneumatic-only mode; (**d**) Extension state of cable-only mode; (**e**) Extension of dual-drive mode.

**Figure 8 biomimetics-08-00614-f008:**
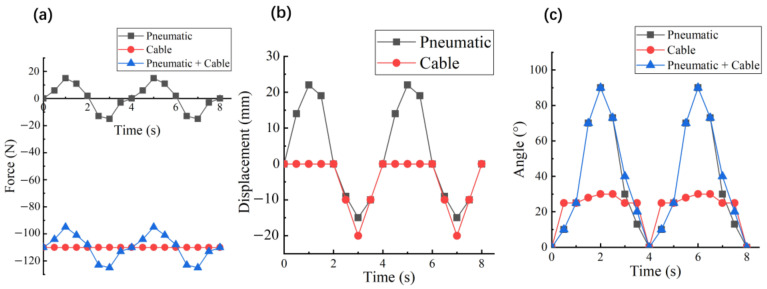
(**a**) Changes in output force within two cycles in three working modes, push force is positive and pull force is negative; (**b**) displacement changes in two cycles under two working modes. The initial displacement is 0, so the wire pull actuator cannot produce positive displacement; (**c**) rotation angle changes in two cycles under three working modes, the initial rotation angle is about 25°.

**Table 1 biomimetics-08-00614-t001:** The working status table of each module under the three working modes.

	Pneumatic Single		Cable Single		Dual-Drive	
High Pressure Pump	on	off	off		on	off
High Pressure Valve	on	off	off		on	off
Vacuum Pump	off	on	off		off	on
Vacuum Valve	off	on	off		off	on
Drain Valve	off		on		off	
Servo Motor	reverse	forward	reverse	forward	reverse	forward

**Table 2 biomimetics-08-00614-t002:** The working status table of each module under the three working modes.

	Pneumatic Single	Cable-Only	Dual Drive
Flexion	Comfort in human–computer interaction.	This action cannot be achieved.	Thrust is provided by pneumatic actuator; the comfort level is the same as pneumatic.
Extension	The maximum pulling force has limitations. Suitable for patients with muscle weakness but not for patients with muscle spasm.	Can provide a large pulling force and is suitable for various hand dysfunctions.	The pulling force is mainly provided by the cable. Can provide a large pulling force and is suitable for various hand dysfunctions.

## Data Availability

All new data generated in this study are reported in the article.

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
