# Peer review of "Soft Finger Rehabilitation Exoskeleton of Biomimetic Dragonfly Abdominal Ventral Muscles: Center Tendon Pneumatic Bellows Actuator"

_biomimetics, 2023, doi:10.3390/biomimetics8080614_

Round 1
Reviewer 1 Report
Comments and Suggestions for Authors
The manuscript describes the combination of a cable driven actuator and a pneumatic actuator. The output of the device is a straightforward combination of the two different mechanisms. The manuscript has a number of significant weaknesses, such as an unclear description of the biomimetic aspect of the work and an unclear justification of the need for the multiple actuation mechanisms. The use of multiple actuation mechanisms makes the controller of the device much more complex, but the manuscript does not describe how this is required for the application.
Comments:
1. Figure 1(b): It would be more instructive if the schematic could show the direction of the muscle contraction so that the readers can understand how the mechanism actually works. Right now, the schematic is just a structure with different colors that doesn’t really show anything about the mechanics, which makes it hard to see the parallels with the proposed actuator designs.
2. Related to the previous point: the way the article is written makes it sound like the only biomimetic aspect of the work is the general concept of a tubular structure. Are there any structural or functional aspects of the actuator mechanism that are similar to that of the dragonfly? The dragonfly changes the bending angle by modifying the muscles in the tail, while the actuator designed in this work only changes from extension to bending based on the
3. The work includes two types of actuators, cable driven and pneumatic. Furthermore, the pneumatic actuator appears to have both positive pressure and negative pressure, which means that there are a total of 3 control signals to operate the actuator. It is important to justify having such a large control system, especially for a wearable device that is supposed to be light. What specific actuation modes does this system enable that are not possible with a simpler actuator. For example, the force profiles in Fig. 6a seem like they would be possible to achieve using only a cable-driven actuator. Since the cable can provide the contractile force, is the vacuum pump necessary to provide contraction using the pneumatics?
4. The introduction mentions that this system is specifically tailored for the needs of wearable rehabilitation devices. However, the requirements for these types of devices are not described, and the suitability for the application are also not described. Is the enabling aspect of the work that the cable can provide large pulling forces while the pneumatic actuator can provide moderate pushing forces?
5. 192: “while ensuring that its size and weight remain within acceptable limits.”
How were these acceptable limits determined?
6. 391: “possesses significant tensile strength”
What is the tensile strength? I can’t seem to find a number regarding this in the manuscript. If this is a really important part of the work, then it would be good to include some data about it in one of the figures.
7. For the bending actuation in Fig 6c, is the bending caused just by constraining the actuator to actuate around a joint rather than linearly? If the actuator itself is not modified to make it a bending actuator, correct?
Reviewer 2 Report
Comments and Suggestions for Authors
Revision of biomimetics-2637867
This work reports on the design and fabrication of a bio-inspired central cable driven pneumatic bellows actuator. From a scientific and technological point of view, the topic is relevant, topical and interesting. However, in order to upgrade the work to scientific, it needs to be improved and deepened in terms of discussion of the results and comparison with the literature in the field. With this in mind, I recommend a Major Revision to be considered for publication in the Bioemimetics Journal.
Comments/Suggestions:
1. The introductory section could be completed with some works on the subject of "dragonfly mimetics", which can be easily found with a quick search in scientific browsers.
2. Discussion of the results needs to be deepened and improved.
3. Please compare your results with the literature and/or commercial systems to better understand the advantages and disadvantages of your system.
4. Minor annotations were added to the PDF of the manuscript.

Comments on the Quality of English LanguageNo comments.
Round 2
Reviewer 1 Report
Comments and Suggestions for Authors
Authors have adequately attempted to address the reviewer comments.
Author Response
Thank for your kindly review and comments.
Reviewer 2 Report
Comments and Suggestions for Authors
Revision of biomimetics-2637867 – Round#2
Comments/Suggestions:
The authors claim to have added content about Dragonfly Bionics, but in fact not a single new reference, scientific or commercial, has been introduced. Please do so.

Author Response
Thank you very much for your comments on our last revision of the manuscript. Although we revised the introduction section, we regret that we neglected to add references.
Thanks to your timely comments, in this revision, we have listed the literature and books that the manuscript referenced last time and this revision in the text About bio robotics, dragonfly-inspired bionic robots, and the basics of introducing arthropods and dragonfly belly rolls.